# Tryptone in Growth Media Enhances *Pseudomonas putida* Biofilm

**DOI:** 10.3390/microorganisms10030618

**Published:** 2022-03-14

**Authors:** Marge Puhm, Hanna Ainelo, Maia Kivisaar, Riho Teras

**Affiliations:** Institute of Molecular and Cell Biology, University of Tartu, 51010 Tartu, Estonia; marge.puhm@gmail.com (M.P.); hannaainelo@gmail.com (H.A.); maia.kivisaar@ut.ee (M.K.)

**Keywords:** *Pseudomonas putida*, biofilm, growth parameters, media components, peptides, Fis

## Abstract

Extracellular factors and growth conditions can affect the formation and development of bacterial biofilms. The biofilm of *Pseudomonas putida* has been studied for decades, but so far, little attention has been paid to the components of the medium that may affect the biofilm development in a closed system. It is known that Fis strongly enhances biofilm in complete LB medium. However, this is not the case in the defined M9 medium, which led us to question why the bacterium behaves differently in these two media. Detailed analysis of the individual medium components revealed that tryptone as the LB proteinaceous component maintains biofilm in its older stages. Although the growth parameters of planktonic cells were similar in the media containing tryptone or an equivalent concentration of amino acids, only the tryptone had a positive effect on the mature biofilm of the wild type strain of *P. putida*. Thus, the peptides in the environment may influence mature biofilm as a structural factor and not only as an energy source. Testing the effect of other biopolymers on biofilm formation showed variable results even for polymers with a similar charge, indicating that biopolymers can affect *P. putida* biofilm through a number of bacterial factors.

## 1. Introduction

Biofilm is usually thought of as mucus-covered sessile bacteria, but it is a more complex bacterial community that acquires new properties that no single cell has. More importantly, biofilm is a dynamic phenotype that changes according to environmental conditions [1,2].

The biofilm of the cosmopolitan bacterium *Pseudomonas putida* has been studied for decades because it is an agriculturally important soil organism that promotes plant growth. *P. putida* protects plants from pathogens by colonising roots and stimulates plant growth by producing plant hormones and mineralising soil phosphorus and metals. Colonisation begins with attachment to plant roots followed by biofilm formation or migration on the roots [3,4,5,6,7]. It is important to note that *P. putida* biofilm developed in closed systems fluctuates under laboratory conditions during the first 24 h. The biofilm increases rapidly during the first 4 to 6 h after inoculation, followed by a decrease to approximately half of the maximum, which can be referred to as mature biofilm [8,9].

Depending on the strain of *P. putida* and the conditions, the biofilm matrix can contain cellulose (enzymes encoded by *bcs* genes), exopolysaccharides A and B (*pea* and *peb* genes), alginate (*alg* genes), DNA, and proteins [8,10,11,12]. The matrix components are substitutable, and some of them are synthesised conditionally. For example, alginate biosynthesis is turned on in water-limiting conditions, and one of the important adhesins, LapF, is expressed in the stationary phase [11,13]. However, when all genes encoding polysaccharides or adhesins are deleted, biofilm is still formed, indicating the flexibility of the biofilm formation [8,12]. On the other hand, the already formed *P. putida* biofilm is sensitive only to certain types of enzymes; for example, DNase does not damage the biofilm, but proteinase K significantly degrades it [8]. It seems that the proteinaceous component of the *P. putida* matrix is essential for the integrity of the biofilm, especially as a structural component. The two surface proteins LapA and LapF are responsible for attachment, initial biofilm formation, and integrity of the mature biofilm. LapA is critical for attachment and biofilm formation of these two adhesins, while LapF is characterised as an adhesin required for cell–cell interaction in a mature biofilm [8,14,15,16].

Bacterial attachment to a surface can be conditionally divided into (i) specific attachment by receptor-ligand binding, or (ii) non-specific attachment where the whole cell binds to a surface. An example of a specific attachment is the *Escherichia coli* Fim pili, which use the adhesin FimH to recognise mannose residues of mucin in the intestinal epithelium. It is also interesting for *E. coli* attachment that shear stress is required for the solid attachment [17,18]. For non-specific attachment, cell surface properties are essential: there is no specific ligand(s), and only the hydrophobicity and electrostatic forces of the cell surface are relevant for adhesion. *P. putida* probably utilises the second type of attachment, as it is not yet known that a specific ligand is required to adhere to a surface.

LapA is the largest protein in *P. putida* (888 kDa), and LapF is the second-largest protein (615 kDa) [19,20,21]. Because Lap proteins are located on the cell surface, both proteins possibly regulate cell surface properties. Both *lapF* expression and cell surface hydrophobicity coincide with a decrease in growth rate and the entry of planktonic cells into the stationary growth phase [14]. In the absence of LapF, the cells no longer became more hydrophobic in the stationary phase, and the cell surface is relatively hydrophilic, similar to the cells in the exponential phase. LapA, on the other hand, is not known to alter cell surface properties [14].

The transcriptional regulation of *lapA* and *lapF* is almost entirely described [13,22,23,24,25], and the post-translational regulation is assumed to be similar to *Pseudomonas fluorescens* [26]. The global regulator Fis has a remarkable role in the complex transcriptional regulation of *lapA* and *lapF* [13,22]. Fis has at least two binding sites in front of the *lapA* gene, which are required to enhance *lapA* transcription. At *lapF*, Fis has one binding site which overlaps with the RpoS-dependent promoter, and Fis binding there represses transcription of *lapF* [13,22].

*P. putida fis*-overexpression strain F15 was used to establish the role of Fis in the regulation of *lapA* and *lapF* expression and biofilm development due to the lethality of the *fis* knock-out mutation for *P. putida*. Moreover, clones with a reorganised genome tended to arise when a downregulated *fis*-expression strain was constructed. Despite Fis’ potential influence on global cell physiology, it had a specific hydrophobicity effect through a binding site in front of the *lapF* gene [3,13,22]. When this binding site was mutated, the surface hydrophobicity of the cells changed similarly to the wild type strain, indicating that the overexpression strain is usable with proper controls [13,22]. *P. putida* growing in LB with the induced expression of *fis* had a positive epistatic effect on mature biofilm, in contrast to the cells with a native amount of Fis, whose biofilm degraded to approximately half-maximum. It is remarkable because Fis effectively represses *lapF* transcription, reducing cell surface hydrophobicity, which should be a prerequisite for biofilm formation [11,14,15].

Under certain conditions, LapF is required for biofilm formation, indicating the critical role of environmental factors. *P. putida* requires LapF adhesin to form a biofilm in a defined M9 medium with glucose or a highly diluted LB medium, but not in the complete LB medium [8,16]. It appears that the components in the medium may determine the ability of *P. putida* to form biofilm. Another contradiction is that the surface of planktonic *P. putida* becomes hydrophobic in the stationary phase and under the influence of stressors; thus, *P. putida* is thought to tend to form a biofilm in response to stress or having a hydrophobic surface [14,27,28]. It is contradicted by studies of the dynamics of *P. putida* biofilm development that show more biofilm at the beginning of active growth, which could reflect the exponential growth of planktonic cells [8,9]. Thus, all these results indicate that the formation of *P. putida* biofilm is more complex and may depend on environmental conditions such as growth media components.

The variants of defined growth medium M9 and complete medium LB are the mainly used media for *P. putida* biofilm formation studies. In this manuscript, we focused on biofilm formation of *P. putida* in the defined M9 supplemented with 0.2% of glucose and 0.2% CAA (M9-0.2CAA) and complete LB media. The main aim was to ascertain why *P. putida* forms biofilm effectively in LB and ineffectively in the M9-0.2CAA medium. Our results indicated that extracellular peptides as architectural factors might have a crucial role in maintaining *P. putida* mature biofilm. 

## 2. Material and Methods

### 2.1. Bacterial Strains and Media 

In the study, *P. putida* wild type strain PSm (Sm^r^) and *fis*-overexpression strain F15 (Gm^r^) were used [3]. Bacteria were grown at 30 °C. Gentamicin at 10 µg/mL or streptomycin at 200 µg/mL were used as antibiotics for inoculum. Biofilms were assessed in media without antibiotics, and 1 mM IPTG was used to overexpress *fis* from the P*_tac_* promoter in *P. putida* F15.

Bacteria were grown in a complete LB medium containing 10 g/L tryptone (LabM, Heywood, UK), 5 g/L yeast extract (LabM, Heywood, UK), and 5 g/L NaCl, or in media based on M9 buffer [29]. All M9 media contained M9 buffer [30], 2.5 mL/L of microelements [31], 2 g/L of glucose, and different amounts of CAA (casamino acid; Difco, Detroit, MI, USA) with tryptophan. The medium M9-0.2CAA contained 2g/L CAA and 0.01 g/L tryptophan, and M9-1.4CAA contained 14 g/L CAA and 0.07 g/L tryptophan. M9-0.2CAA was supplemented with tryptone (LabM, Heywood, UK), gelatin peptone (Sigma-Aldrich, St. Louis, MO, USA), soybean peptone (Acros Organics, Geel, Belgium), bacteriological peptone (Amresco, Solon, OH, USA), or bacteriological tryptone (Amresco, Solon, OH, USA) at 10g/L. Alternatively, M9-0.2CAA was amended with polypeptides (0.4 mg/mL poly-L-lysine (1000–5000 Da); 0.4 mg/mL poly-L-glutamic acid (1500–5500 Da); and 0.4 mg/mL poly-DL-alanine (1000–5000 Da)) or biopolymers, with sodium salt of carboxyl methylcellulose (Sigma-Aldrich, St. Louis, MO, USA), PGA (poly-galacturonic acid), and salmon sperm DNA or BSA (bovine serum albumin) at 0.4 g/L. 

To identify components needed for the biofilm-enhancing effect of Fis, M9-0.2CAA was amended with LB components one by one: 10 g/L tryptone (LabM, Heywood, UK), 5 g/L yeast extract (LabM, Heywood, UK), and 5 g/L NaCl and vice versa. LB was amended with components of M9-0.2CAA one by one: M9 buffer, 2.5 mL/L of microelements, 2 g/L of glucose, and 2 g/L of CAA and 0.01 g/L tryptophan. 

### 2.2. Assessment of Biofilm and Growth Parameters of Planktic Cells

Biofilm formation was monitored in flat-bottom microtiter plates (Greiner Bio-One International; 96-well cell culture plate Cat-No 655180) according to the Fletcher method [32], with some modifications [3]. LB medium with digested peptides was prepared as follows: LB was treated with 1 U/mL proteinase K at 37 °C for 2 h. Proteinase K was inactivated by heating the medium at 121 °C for 20 min. Residual proteinase K activity was assessed by degradation of 1 mg/mL BSA in treated medium maintained at 37 °C with BSA for 2 h. No proteinase K treatment was performed for the controls.

The method of Lopez-Sanchez et al. [9] was used to assess biofilm formation over 24 h. In short, the procedure is based on the decimal dilution of inoculum, which delays biofilm formation. Every subsequent dilution gives so-called younger biofilm, and after intercalation of biofilm yields a planktic growth curve, it is possible to assess the biofilm dynamics [9].

The fresh medium was inoculated overnight with bacterial culture to an initial calculated density of 0.1 (A_580_). From the first inoculation, 10-fold dilutions up to 10^8^-fold were made. To measure biofilm with sufficient frequency over time, *P. putida* PSm biofilm at 6, 13, 14, 15, and 24 h and F15 biofilm at 6, 16, 17, 18, 19, and 24 h were evaluated. Assessing the growth of planktic cells from the same dilutions, the biofilm was intercalated to the growth curve based on the shift in the exponential growth initiation point on the growth curves of the decimal diluted cells.

To determine growth parameters, *P. putida* was grown overnight in the appropriate medium for inoculum. The fresh medium was inoculated overnight with bacterial culture to an initial calculated density of 0.1 (A_580_). *P. putida* was grown in flat-bottom microtiter plates (Greiner Bio-One International; 96-well cell culture plate Cat-No 655185) to a final volume of 150 µL per well at 30 °C. If necessary, 1 mM IPTG was added. The optical density of the bacterial culture was measured every 7 min for 24 h. Bacterial growth rate (µ) and maximal absorbance (maxOD) values were estimated from the growth curves using the Gompertz model [33]. The mean of growth rate and maxOD were calculated from 8 individual parallels.

### 2.3. Western Immunoblot Assay

Western immunoblot analysis was carried out to detect the amount of Fis from the crude lysates of *P. putida* F15. Crude cell lysates were prepared from 18-h-old cells grown in 5 mL LB, M9-0.2, M9-1.4, or M9-Trp media. Then 1 mM IPTG was used in media if it was needed. The cells were collected by centrifugation and suspended in the RIPA buffer described previously [8]. 

The total amount of protein in the cleared supernatant was measured spectrophotometrically by the content of tryptophan [34]. For Western blotting, 10 μg of total protein was loaded per well and separated by 12% Tricine-SDS-PAGE [35]. Fis was detected by mouse anti-Fis as previously described [8].

### 2.4. Statistical Analysis of the Results

Factorial ANOVA was used to assess the variability of data in experiments. The multiple comparisons of means were conducted following Bonferroni’s test for unequal *n*. For statistical tests, the significance level was set at *p* < 0.05. Homogenous groups with the same letter indicate similarity of the arithmetic mean (*p* ≥ 0.05). Using the General Linear Model, the dynamics of *P. putida* biofilm grown in all four media were compared. The "medium "and "IPTG" were set as categorical factors and "time" as a continuous factor. Spearman correlation analysis was used for growth parameters, biofilm, IPTG, and tryptone for *P. putida* strains PSm and F15. Tryptone was defined as "1" for LB and M9-tryptone media and "0" for M9-0.2CAA and M9-1.4CAA media. The average value of every media biofilm was used for analysis, as the number of measurements was different for biofilm and growth parameters of planktic cells. The calculations were performed using Statistica 13 software.

## 3. Results

The biofilm of *Pseudomonas* is a dynamic feature and can depend on extracellular factors and the physiological state of bacteria [8,36]. We have studied *P. putida* biofilm formation and observed that the growth medium and the age of bacterial culture strongly affect biofilm formation [3]. The current study aimed to determine extracellular factors that are involved in *P. putida* biofilm formation. We used *P. putida* wild type strain PSm and an artificial strain F15, a *fis*-overexpression derivate of PSm [3]. The biofilm of both strains, PSm and F15, was assessed in different media to ascertain extracellular factors that affect its formation.

The biofilm was assessed in microtiter plates containing LB or M9 medium with 0.2% glucose and 0.2% casamino acids with tryptophan (M9-0.2CAA) in the absence or presence of 1 mM IPTG. The IPTG was used to induce the *fis* gene from the P*_tac_* promoter in F15. As expected, *P. putida* strains’ biofilm grown without IPTG was 1.6–1.8 times higher in LB medium than M9-0.2CAA medium (Figure 1A). However, surprisingly, overexpression of *fis* by IPTG did not enhance *P. putida* biofilm in the M9-0.2CAA medium (Figure 1A). The overexpression of *fis* by 1 mM IPTG was confirmed by immunoblot analysis (Figure 1B). Thus, the possibility that F15 has lost the enhanced biofilm in M9-0.2CAA (or M9-1.4CAA, see below) due to the inability to induce *fis* expression in F15 by IPTG was ruled out. Therefore, we focused on the possibility that extracellular factors modulate Fis’ capacity to enhance biofilm formation.

### 3.1. Biofilm in Mixed LB and M9-0.2CAA Media

There are two possibilities for how the growth media components could affect the biofilm: either the M9 components inhibit or the LB components promote biofilm formation, especially the emergence of the enhanced biofilm of F15 by IPTG. To identify the extracellular factor that could affect biofilm formation, we mixed components of LB and M9-0.2CAA media. Each medium was supplemented one by one with individual components of the other medium in their original concentrations. Tryptone, yeast extract, and NaCl were added to M9-0.2CAA; and phosphate buffer, glucose, CAA, and microelements were added to LB (Figure 2).

The supplementation of M9-0.2CAA by tryptone increased the biofilm of *P. putida* wild type strain PSm by 1.3 times (*p* = 0.029) and reproduced the *fis*-overexpression effect on F15 biofilm, as seen when bacteria were grown in LB (Figure 2A). Notably, in the presence of IPTG, the biofilm of F15 was increased 2.9 times (*p* < 0.0001) compared to the absence of IPTG in the M9-0.2CAA+tryptone medium (Figure 2A). Yeast extract also increased F15 biofilm in the *fis*-overexpression background, but the effect was weaker (1.6-fold increase upon IPTG addition, *p* < 0.0001), and the yeast extract did not affect the biofilm of PSm (Figure 2A). Additional NaCl in the M9-0.2CAA medium did not have a remarkable effect on the biofilm of PSm.

The presence of M9-0.2CAA components in LB did not significantly affect the biofilm of *P. putida* wild type strain PSm. However, CAA and microelements reduced the biofilm of PSm 1.3 times compared to the biofilm measured from LB without supplementation, but the effect was statistically insignificant (Figure 2B). Despite M9-0.2CAA components in the LB medium, overexpression of *fis* increased the biofilm of F15 in the LB medium (Figure 2B). 

In conclusion, adding tryptone to the M9-0.2CAA medium increased the biofilm of *P. putida* wild type strain PSm and restored the biofilm enhancement by *fis*-overexpression in the F15 strain to similar levels as observed in LB (Figure 1) [3]. The M9-0.2CAA components in LB did not meaningfully affect *P. putida* biofilm. Thus, it seems that the proteinaceous factor in media increases *P. putida* biofilm. 

### 3.2. P. putida Biofilm in Proteinase K Treated LB Medium

LB contains proteinaceous components from tryptone and yeast extract. To investigate whether these are responsible for Fis-mediated biofilm enhancement, LB was treated with proteinase K. Before inoculation of the treated medium with cells, proteinase K was thermally inactivated to prevent the action of protease K on the forming biofilm (Figure 3). Both studied strains of *P. putida* formed less biofilm in the proteinase K-treated LB medium than without the treatment (Figure 3A). Moreover, the Fis-enhancing effect on F15 biofilm was missing in the proteinase K-treated LB (Figure 3A). These results indicated that the proteinaceous component in the LB medium is responsible for enhanced biofilm formation when *fis* is overexpressed.

### 3.3. Biofilm and Growth of P. putida in M9 Medium with Different Concentrations of CAA

Tryptone is a trypsin hydrolysate of casein or some other protein and contains oligopeptides in different lengths and smaller amounts of amino acids [37,38]. Therefore, tryptone can affect *P. putida* growth as an additional C- and N-source. To distinguish whether the effect of tryptone relies on peptides in particular or the extra nutrients it provides, we used an equivalent amount of casamino acid + tryptophan as a control (Figure 4). Casamino acid (CAA) is an acid hydrolysate of casein, mainly containing amino acids, except for tryptophan and a minor amount of short peptides [38,39]. Roughly 1.4% of CAA+Trp should provide the same amount of amino acids as tryptone of LabM (Heywood, UK) added to LB [40,41].

Biofilm was measured in M9 0.2% glucose medium amended with 0.05%, 0.1%, 0.2%, 0.4%, 0.8%, 1.4%, and 2% of CAA+Trp for 24 h (Figure 4). The amount of CAA+Trp in the M9 medium did not affect the biofilm of the wild type strain PSm (Figure 4). More importantly, the F15 had no biofilm-enhancing effect by IPTG at any concentration of CAA in M9 (Figure 4). 

Thus, taking together, the proteinaceous component per se, as peptides in the medium, enhances the biofilm formation of *P. putida.*

### 3.4. The Dependence of P. putida Biofilm on Growth Parameters

Next, we asked whether it is possible to evaluate biofilm using planktonic bacteria’s growth parameters (e.g., growth rate and maximum OD). The growth parameters vary broadly depending on the bacteria’s growth medium and genetic background. The growth rate reflects the speed of biomass formation from the nutrients in the medium, and maximum OD (maxOD) indicates the conversion efficiency of nutrients to biomass. In the current study, we were interested in whether the presence of tryptone could influence growth rate, maximum OD, or biofilm (Appendix A and Table 1). The growth rates and maxOD of bacteria grown in M9-1.4CAA and M9-0.2CAA+tryptone were of particular interest. These media contained similar amounts of amino acids, and therefore we expected that the growth parameters could be more similar to LB than to M9-0.2CAA. 

Indeed, the growth rate and maxOD of the wild type strain PSm depended on amino acid concentration in the growth medium (Appendix A). The presence of tryptone or extra CAA in the medium (LB, M9-0.2CAA+tryptone and M9-1.4CAA) increased maxOD and decreased the growth rate of PSm, differing remarkably from growth parameters in M9-0.2CAA (Appendix A). The effect of IPTG on the growth rate or maxOD of *P. putida* wild type strain PSm was absent in most cases; only the growth rate in M9-0.2CAA+tryptone was slightly lower in the presence of IPTG (Appendix A). The media’s high amino acids component affected the maxOD and growth rate of F15 differently, but most importantly, IPTG decreased the growth rate and maxOD of F15 in all media, except in M9-1.4CAA, where the growth rate was not affected by IPTG (Figure 4). The growth rates and maxOD of F15 grown in M9-1.4CAA were more similar to the growth parameters obtained in M9-0.2CAA+tryptone than M9-0.2CAA (Appendix A). 

Thus, knowing that the growth of planktic bacteria is dependent on the presence of a proteinaceous factor on media, it was essential to understand how this affects biofilm formation (Table 1). The most substantial correlation with the biofilm of *P. putida* wild type strain PSm appeared with the presence of tryptone, indicating that the presence of tryptone is linked to more biofilm. At the same time, tryptone is weakly correlated with growth rate and maxOD. These results indicated that tryptone has a positive effect on biofilm per se and excluded a possible effect of tryptone as the compound of extra C- and N-sources (Table 1). However, the PSm growth rate was negatively correlated with both biofilm and maxOD; and the maxOD and biofilm were positively correlated (Table 1). Thus, when *P. putida* PSm grows slower but can produce more biomass, it probably forms more biofilm. As expected, the IPTG correlation with growth rate, maxOD and biofilm of *P. putida* PSm was insignificant, indicating that IPTG does not influence the growth or life form of wild type strain (Table 1).

Similarly to *P. putida* wild type strain PSm, the most significant correlation was ascertained between tryptone and biofilm of F15 (Table 1). While the presence of tryptone in the medium affected the F15 biofilm positively, IPTG had a similar positive effect (Table 1). However, the growth rates of F15 and maxOD were negatively correlated, which was also the case with the wild type strain PSm (Table 1). Unlike the wild type strain, the addition of IPTG to the growth medium affected the maxOD and growth rate of F15 negatively (Table 1).

Taken together, the growth parameters of *P. putida* planktic cells did not describe entirely the ability of *P. putida* to form a biofilm, but tryptone did. The 24-h-old biofilm depends on extracellular tryptone that probably is not used only for metabolism. 

### 3.5. Dynamics of P. putida Biofilm

The growth rate in combination with IPTG may affect biofilm formation; for example, the presence of IPTG in the F15 growth medium may delay biofilm formation as it decreases the growth rate of F15 (Table 1). Therefore the 24-h-old biofilms of F15 grown with and without IPTG may not be comparable to each other. Thus, we assessed the biofilm formation over 24 h by the method of Lopez-Sanchez et al. [9]. The procedure is based on the decimal dilution of inoculum, which delays biofilm formation. Every subsequent dilution gives so-called younger biofilm, and after intercalation of biofilm results in the planktic growth curve, it is possible to assess the biofilm dynamics [9].

Using the General Linear Model, the dynamics of PSm biofilm grown in all four media were compared. The "medium "and" IPTG "were set as categorical factors and "time" as a continuous factor. Firstly, from the three main effects, only “medium“ had a statistically significant impact on PSm biofilm (p = 7.7 × 10^−14^), while “IPTG“ (*p* = 0.498) and “time“ (*p* = 0.332) were insignificant. Thus, the most critical differences in wild type strain PSm biofilm were caused by media but not by IPTG or time (Figure 5). Secondly, from the second level of interactions, only "medium+time" had a statistically significant effect on the biofilm of PSm (*p* = 1.8 × 10^−8^); the rest of the second level or the third level of interactions were statistically insignificant. Consequently, PSm biofilm changes over time in the evaluated medium (Figure 5). According to the follow-up Bonferroni test, biofilms formed in media containing tryptone (LB and M9-tryptone) compared to biofilms formed in media without tryptone (M9-0.2CAA and M9-1.4CAA) were significantly different (any media with tryptone versus any media without tryptone *p* < 0.0001; LB vs. M9-0.2CAA+tryptone *p* = 0.160; M9-0.2CAA vs. M0-1.4CAA *p* = 1).

The quick formation of biofilm and the dispersion of young biofilm were only observable in the media without tryptone (M9-0.2CAA and M9-1.4CAA; Figure 5A,B). However, the early biofilm was not measurable from media containing tryptone for technical reasons. In all cases, bacteria formed stable biofilm after 8 h (Figure 5); thereby, 24-h-old biofilm reflects stable, mature biofilm in all used conditions.

Additionally, the average of mature biofilms was calculated from the biofilms obtained in 8 to 24 h and compared by multifactorial ANOVA (Figure 5). The effect of tryptone was evident similarly to the previous analysis; tryptone in media (LB and M9-tryptone) increased the average biofilm of PSm compared to the average biofilm formed in media without tryptone (M9-0.2CAA and M9-1.4CAA; Figure 5E). Indeed, despite the same amount of C- and energy sources in media (compare M9-1.4CAA and M9-tryptone), wild type *P. putida* formed a more mature biofilm in the presence of tryptone. 

The biofilm of *P. putida* strain F15 was similarly assessed to the wild type strain PSm. All main effects had a statistically significant impact on the Fis biofilm (medium: *p* = 4.0 × 10^−10^; IPTG: *p* = 7.4 × 10^−10^; time: *p* = 2.3 × 10^−8^), indicating that the biofilm of F15 changes over time in the presence of IPTG (by an elevated amount of Fis; Figure 6). The second level interaction "IPTG+medium" (*p* = 1.3 × 10^−6^) and "medium+time" (*p* = 2.8 × 10^−5^) had a statistically significant impact on the biofilm of F15, showing that the effect of Fis on F15 biofilm depends on the medium where biofilm formed and the development of F15 biofilm is time-dependent in different media (Figure 6). The rest of the interactions were statistically insignificant. 

Thus, the effect of *fis*-overexpression on F15 mature biofilm was prominent in the presence of tryptone (Figure 6C,D) but not in the absence of tryptone (Figure 5A,B). In addition, the young F15 biofilm lacked the effect of *fis*-overexpression in tryptone-containing medium (Figure 5C,D), being consistent with previously obtained F15 results in LB [8].

Evaluation of F15 mature biofilm in a biofilm dynamics study, with the mature biofilms calculated as averages of the biofilms obtained in 8 to 24 h, confirmed the requirement of tryptone for Fis-enhanced biofilm (Figure 1, Figure 2, and Figure 6E). Although much less pronounced, a similar effect was seen in the M9-1.4CAA medium, probably due to the small number of peptides that CAA could contain (Figure 6E).

In conclusion, tryptone in media with similar C- and energy sources promotes the staying of *P. putida* cells in the mature biofilm. Thus, under the conditions evaluated, the extracellular proteinaceous component appears to be another essential factor for biofilm formation, not just a source of carbon and energy.

### 3.6. Involvement of Polypeptides in the Biofilm of P. putida

Knowing that the presence of peptides in the M9-0.2CAA medium increases the mature biofilm, the question arose as to whether adding a different mix of peptides could restore the effect of tryptone used in our studies (purchased from LabM, Heywood, UK). Thus, biofilm formation was monitored in M9-0.2CAA supplemented with peptones and tryptone other than LabM (Heywood, UK), such as gelatin peptone (Sigma-Aldrich, St. Louis, MO, USA), soy peptone (Acros Organics, Geel Belgium), bacteriological peptone (Amresco, Solon, OH, USA), and bacteriological tryptone (Amresco, Solon, OH, USA). Because the biofilm-enhancing effect of Fis was readily detectable, only the *fis*-overexpression strain F15 was studied. 

Surprisingly, the biofilm-enhancing effect of Fis was more variable when using miscellaneous peptones. The F15 biofilm was independent of the level of expression of *fis* when gelatin peptone (Sigma-Aldrich, St. Louis, MO, USA) was used (Appendix A). The *fis*-overexpression resulted in a 1.7-fold higher biofilm in the presence of soybean peptone (Acros Organics, Geel, Belgium) and a 2.3-fold higher biofilm when bacteriological peptone (Amresco, Solon, OH, USA) was added to the medium compared to the corresponding biofilm formed without the overexpression of *fis* (Appendix A). However, Amresco (Solon, OH, USA) tryptone amplified the biofilm in the presence of IPTG 3.9-fold compared to the biofilm formed without the addition of IPTG (Appendix A). Thus, these results suggest that some peptides may be more suitable for a mature biofilm than others. When comparing peptides obtained by treating pepsin and trypsin, these peptides differ in their terminal amino acid. Pepsin treatment yields peptides with hydrophobic amino acids at the one end, while trypsin treatment yields peptides with a positively charged amino acid [42,43,44]. This knowledge encouraged us to replace tryptone in M9-0.2CAA+tryptone with differently charged peptides. We were interested in whether the charge of the peptides could be the cause of the enhanced biofilm. Therefore, three types of polypeptides were added to the M9-0.2CAA medium: positively charged poly-L-lysine (pK), negatively charged poly-L-glutamate (pE), and uncharged poly-DL-alanine (pA) at final concentrations 0.4 mg/mL. Initially, the poly-L-lysine in concentrations 0.13, 0.2, 0.4, and 1 mg/mL was used to find the effect of the polypeptide on the biofilm (data not shown). Seeing that a concentration of 0.4 mg/mL enhances the F15 biofilm the most stably when *fis* was overexpressed, the same concentration was used for the rest of the polypeptides. 

Of all the polypeptides used, only pK showed a prominent effect on the 24-h-old biofilm (Figure 7A). Namely, the enhancing effect of Fis on the biofilm in the M9-0.2CAA medium was observed only in the presence of poly-lysine. However, pK had no statistically significant impact on the wild type strain PSm of *P. putida* (Figure 7A). In addition to poly-lysine, F15 also had more biofilm in the presence of poly-glutamate, although without the Fis-effect (Figure 7A). Thus, positively charged poly-lysine appeared to facilitate the biofilm when the amount of Fis was increased by IPTG; at the same time, negatively charged poly-glutamate and uncharged poly-alanine did not. 

As the next step of our study, poly-L-lysine was replaced with spermine, a positively charged polyamine, in the M9-0.2CAA medium to examine the effect of positively charged extracellular factors on the F15 biofilm. Spermine elicited a positive effect of *fis*-overexpression on the F15 biofilm, similar to that observed in LB, M9-0.2CAA+tryptone, and M9-0.2CAA+pK medium (Figure 6E and Figure 7A). However, in contrast to pK, spermine has a mild, statistically insignificant negative effect on the wild type strain PSm biofilm (Figure 7A). These results indicated that tryptone could be substitutable for a positively charged compound as poly-lysine or spermine for enhancing *P. putida* F15 biofilm formation. 

### 3.7. The Biofilm of P. putida in Media Amended with Biopolymers

Inspired by the effect of positively charged peptides, we asked whether other possible rhizosphere biopolymers could affect the *P. putida* biofilm. Due to rhizosphere factors, especially in the vicinity of old parts of the roots, *P. putida* could be exposed to many biopolymers, such as polysaccharides, DNA, and proteins, which are released into the environment as a result of the breakdown of plant cells. Thus, water-soluble cellulose sodium salt (negatively charged), polygalacturonic acid (PGA), DNA (salmon sperm DNA), and BSA (bovine serum albumin) were added to M9-0.2CAA, and *P. putida* biofilm was measured 24 h after inoculation. 

The water-soluble cellulose in M9-0.2CAA significantly increased the biofilm of both strains, PSm and F15 (Figure 7B). At the same time, no biofilm enhancement by Fis was seen, indicating that cellulose has an overall positive effect on biofilm (Figure 7B, compare F15 biofilm with and without IPTG). 

Similar to cellulose, the addition of DNA in the M9-0.2CAA medium produced more biofilm than without the addition of DNA (Figure 7B). Compared to the effect of cellulose, the impact of extracellular DNA on the biofilm was negligible, and the impact of PGA was absent (Figure 7B). In contrast, the addition of BSA to the M9-0.2CAA medium was the only biopolymer that elicited a Fis-dependent positive effect on the F15 biofilm. However, the impact of BSA on the wild type biofilm was negligible (Figure 7B). Thus, negatively charged biopolymers, such as cellulose and extracellular DNA, may also increase the *P. putida* biofilm, but this does not appear to be Fis-dependent. 

In conclusion, the biofilm of *Pseudomonas putida* is dependent on extracellular biopolymers. Cellulose, DNA, proteins, and peptides all appear to affect biofilm positively, with various mechanisms involved in enhancement. Therefore, it is essential to consider the content of biopolymers in laboratory media when measuring *P. putida* biofilm, as this may significantly affect the results.

## 4. Discussion

Bacterial biofilm formation is a highly complex feature and can be influenced by various factors, both intra- and extracellular. Biofilm formation is commonly associated with starvation and stressors. Indeed, stress, especially toxic substances, promotes cell aggregation to collectively reduce the surface area exposed to the stressors. Under stress conditions (stationary phase or exposure to harmful substances) and aggregation, the cell surface becomes more hydrophobic. Thus, it is concluded that stress conditions promote biofilm formation. We have noticed two inconsistent aspects with the general theory of biofilm formation. First, *P. putida* biofilm is favoured in the early stages of its appearance, when the environment should contain plenty of nutrients. Conversely, as the nutrients are depleted in a closed system, the biofilm is reduced by dispersion [8,15,24]. Second, *P. putida* can form biofilm efficiently when the cell surface is relatively hydrophilic. The need for some factors, such as LapF, is conditional for biofilm formation depending on growth media [8,14,16]. 

The results of measuring *P. putida* biofilm in LB and M9-0.2CAA were surprising (Figure 1). Fis enhanced biofilm production only in one medium, and this raised the question, why. One possible explanation could be that biofilm depends on the amount of C- and energy source as a complex physiological outcome. 

The biofilm is a collection of cells in a gel matrix, which provides homeostatic conditions reducing stress and thereby energy loss for the bacterial population. The biofilm matrix may consist of different components, but there are certainly biopolymers explicitly produced for the matrix in addition to water. The production of biopolymers is energy-intensive. For this reason alone, it can be expected that biofilm will grow on a nutrient-rich medium and that little development will be possible under conditions where the source of C- and energy is limiting. In other words, to form a biofilm, the bacteria must have sufficient nutrients. For example, experiments with *Pseudomonas fluorescens* showed that the bacterium forms a sufficiently dense biofilm on an optimal glucose medium, 30 mg/L of glucose. At lower (15 mg/L) and higher glucose concentrations (45 mg/L), less biofilm was formed [33]. 

The biofilm can be treated as the sum of the cells and the matrix surrounding them. A decrease in either component reduces the biofilm, indicating less favourable conditions for the biofilm. Under the conditions studied, the *P. putida* biofilm was maximal when bacteria were grown for about six hours, and then the amount of biofilm started to decrease (Figure 5 and Figure 6). These results suggest that the *P. putida* biofilm is rather a welfare phenotype; when unfavourable conditions occur, the biofilm diminishes, either through cell departure or matrix depletion. Thus, the age of the biofilm must be considered when assessing the relationship between C- and energy source and biofilm. Examination of the biofilm dynamics revealed that regardless of the strain studied, *P. putida* formed biofilm efficiently after attachment, followed by a half-maximal degradation of the biofilm (Figure 5A,B and Figure 6A,B). These results are in good agreement with previous results [8,15,24]. In other words, *P. putida* forms a biofilm better in the presence of optimal amounts of nutrients. The lack of nutrients signals the initial biofilm dispersion, i.e., the resumption of mobility in some cells, which would essentially mean finding a more prosperous environment. The biofilm dispersion is estimated to occur at about the same time that the planktonic cells enter the stationary phase, suggesting that the *P. putida* biofilm begins to degrade as nutrients are depleted (Figure 5 and Figure 6). Planktic cells have a larger surface area exposed to the environment than those aggregated in biofilm. This would likely give them a growth advantage and explain the observed behaviour during nutrient deficiency.

However, the older or mature biofilm, measured from 24-h-old cells, which was estimated to coincide with the stationary phase of planktonic cells, was more dependent on the extracellular component tryptone. Both studied strains maintained more mature biofilm in media supplemented with tryptone than media without tryptone (Figure 1, Figure 2, Figure 4, and Figure 5). When tryptone was replaced with an equivalent amount of amino acids, the growth parameters (growth rate and maxOD) of planktonic cells were similar. Still, the medium’s equal amount of amino acids did not provide a more substantial biofilm in the mature phase (Appendix A, Figure 5E, and Figure 6E).

Certainly, planktonic *P. putida* catabolises tryptone, but our results suggest that *P. putida* may use peptides for biofilm integrity as a structural factor. The planktic population grew to a similar maxOD in M9-1.4CAA, M9-0.2CAA+tryptone, and LB (Appendix A), indicating that tryptone is efficiently metabolised in cells. All these media contained similar amounts of amino acids; additionally, M9-1.4CAA and M9-0.2CAA+tryptone had the same amount of glucose and microelements. However, cells formed a higher amount of biofilm in a medium containing tryptone than M9-0.2CAA or M9-1.4CAA, suggesting that sessile cells use tryptone not only for catabolism but also likely as a structural element to build a biofilm. Similarly, the reduced biofilm formation in the protease K-treated LB medium indicates that the critical factor for biofilm is not free amino acids but peptides (Figure 3A). Moreover, BSA, poly-lysine, and spermine in the M9-0.2CAA medium also positively affected the *P. putida* F15 biofilm during overexpression of *fis* (Figure 3 and Figure 7A). Hence, the proteinaceous component may act as a structural factor in the *P. putida* biofilm, likely stabilising the biofilm matrix.

It is important to emphasise that tryptone (peptides) increases the biofilm of the wild type *P. putida* strain PSm and the *fis*-overexpression strain F15 (Figure 2A, Figure 3A, Figure 5E, and Figure 6E, Table 1). However, the effect of positively charged peptides on the biofilm was not apparent. When tryptone was replaced by poly-L-lysine or spermine, the biofilm improved only in the case of *fis*-overexpression (Figure 7A). Poly-lysine may not be the best choice for studying mild changes in the wild type strain of *P. putida*, as cationic peptides are known to degrade biofilms [45,46,47]. Thus, possibly the harmful effects of the cationic peptide overshadowed the enhancing impact for biofilm, and the actual peptide sequence, if any, that would amplify the biofilm of *P. putida* remains to be elucidated. The negatively charged cellulose and even extracellular DNA used in the study promoted a mature biofilm in the M9-0.2CAA medium (Figure 7B). However, PGA, which also had a negative charge, did not affect the formation of *P. putida* biofilm (Figure 7B). Consequently, the effect of biopolymers on biofilm relies on more intricate properties than just their charge. 

Since the effect of tryptone on the *P. putida* biofilm is particularly pronounced in cells with elevated Fis, there must be a Fis-controlled factor that mediates the impact of the peptides on the biofilm. It is likely that one of these proteins, LapA or LapF, could be involved in biofilm formation via an extracellular peptide. Fis is known to enhance *lapA* expression and inhibit *lapF* expression [19,20], leaving the surface of planktonic cells relatively hydrophilic even in the stationary phase compared to wild type PSm [14]. Therefore, our subsequent studies are focused on examining the possibility that LapA and LapF could be involved in forming the peptide-enhanced biofilm.

## 5. Conclusions

This study aimed to investigate why *Pseudomonas putida* forms biofilms differently in growth media and whether peptides in the form of tryptone affect the biofilm more than the C-source. It was ascertained that the *Pseudomonas putida* biofilm was dependent on different types of extracellular biopolymers (cellulose, DNA, proteins, and peptides), all of which appear to have a positive effect on the biofilm. Peptides were found to be the most intriguing, as they were crucial for biofilm enhancement by Fis. Therefore, it is essential to consider the content of biopolymers in laboratory media when measuring *P. putida* biofilm, as this may significantly affect the results. 

## Figures and Tables

**Figure 1 microorganisms-10-00618-f001:**
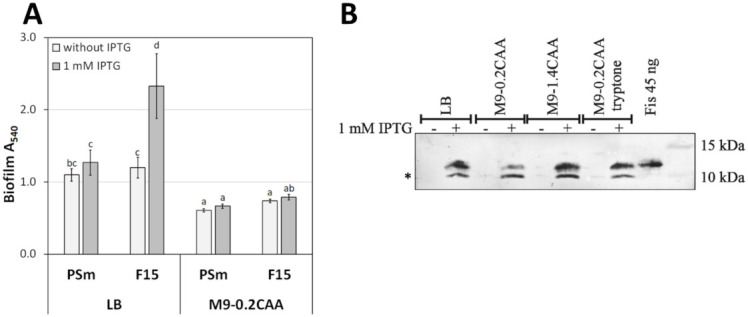
The biofilm of *P. putida* and *fis*-overexpression in strain F15. (**A**) The biofilm of *P. putida* strains PSm (wt) and F15 (*fis*-overexpression strain) grown in the LB and M9-0.2CAA media in the presence or absence of 1 mM IPTG. The arithmetical averages of at least two independent measurement sets are shown. Brackets show the 95% confidence intervals, and the homogenous groups are shown above the columns by the lower-case letters. Identical letters denote non-significant differences (*p* ≥ 0.05) between averages of biofilm. (**B**) Overexpression of *fis* in *P. putida* F15 by immunoblotting using monoclonal anti-Fis antibodies. A total of 10 μg of total protein obtained from *P. putida* F15 grown in LB, M9-0.2CAA, M9-1.4CAA, or M9-0.2CAA+tryptone media was loaded on a 12% SDS-PAA gel. The supplementation of 1 mM IPTG is shown by "+" above the lane. A total of 45 nanograms of purified Fis (6His) was used as a positive control. An asterisk indicates proteolytically cleaved Fis. Protein marker band sizes are indicated.

**Figure 2 microorganisms-10-00618-f002:**
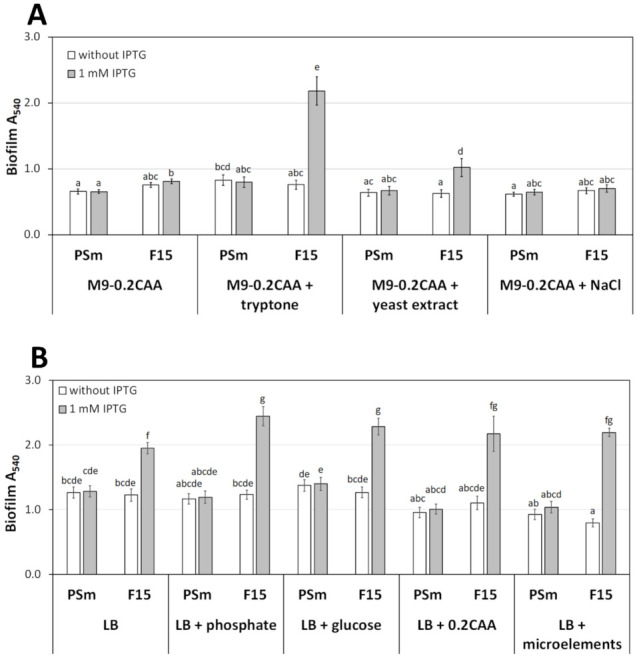
The biofilm of *P. putida* strains PSm and F15 in media supplemented with individual components. (**A**) Biofilm formation in M9-0.2CAA supplemented with single components of LB, and (**B**) biofilm formation in LB medium supplemented with single components of M9-0.2CAA. The arithmetical averages of at least four independent measurement sets are shown. Brackets show the 95% confidence intervals, and the homogenous groups are shown above columns by the lower-case letters. Identical letters denote non-significant differences (*p* ≥ 0.05) between averages of biofilm.

**Figure 3 microorganisms-10-00618-f003:**
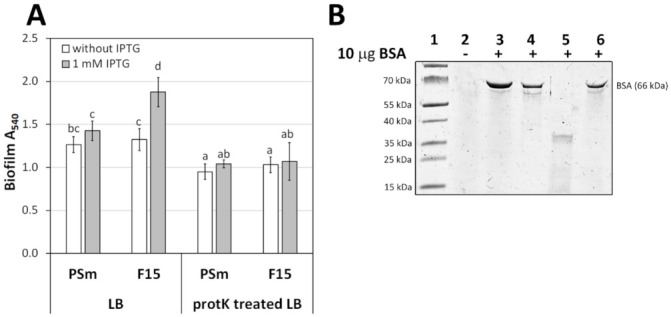
The effect of LB proteinaceous component for *P. putida* biofilm. (**A**) The biofilm of *P. putida* strains PSm and F15 in LB medium and the proteinase K-treated LB medium. Biofilm of *P. putida* strains. The arithmetical averages are shown. Brackets show the 95% confidence intervals, and the homogenous groups are shown above columns by the lower-case letters. Identical letters denote non-significant differences (*p* ≥ 0.05) between averages of biofilm. (**B**) SDS-gel electrophoresis LB medium supplemented with BSA. Lane 1, protein marker; lane 2, 10 μL of LB medium; lane 3, 10 μg of BSA; lane 4, LB medium + BSA without proteinase K treatment; lane 5, LB medium + BSA after 2 h proteinase K treatment; lane 6, proteinase K-treated LB medium amended with 10 μg of BSA after incubation in 121 °C and incubated at 37 °C for 2 h.

**Figure 4 microorganisms-10-00618-f004:**
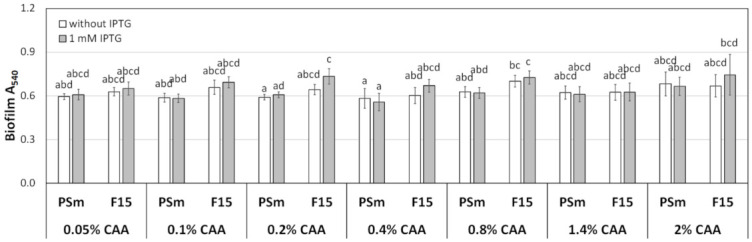
The biofilm of *P. putida* strains PSm and F15 in M9 media was amended with different amounts of CAA. The arithmetical averages of at least two independent measurement sets are shown. Brackets show the 95% confidence intervals, and the homogenous groups are shown above the columns by the lower-case letters. Identical letters denote non-significant differences (*p* ≥ 0.05) between averages of biofilm.

**Figure 5 microorganisms-10-00618-f005:**
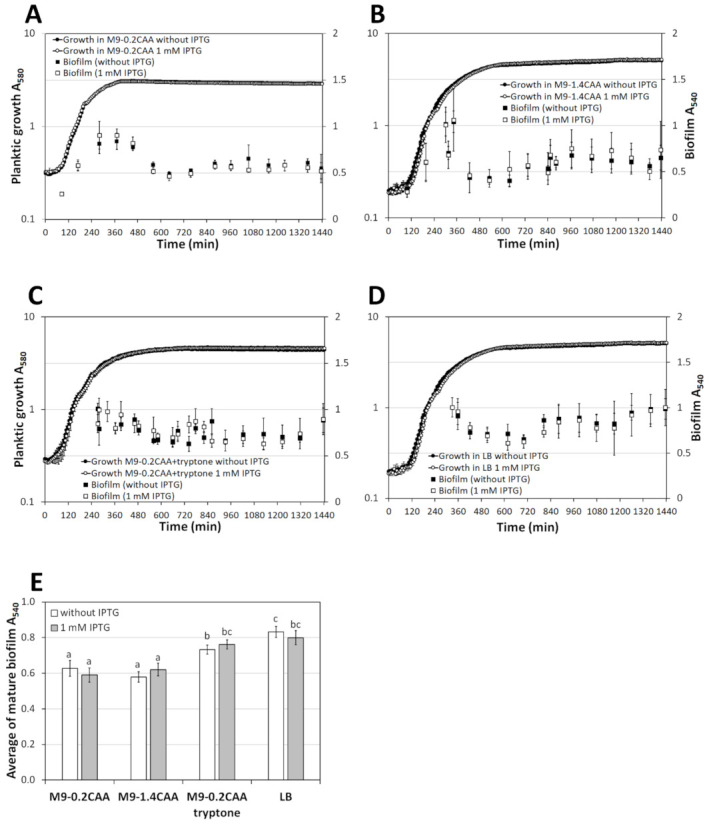
The biofilm dynamics and growth curves of *P. putida* wild type strain PSm grown in (**A**) M9-0.2CAA, (**B**) M9-1.4CAA, (**C**) M9-0.2CAA+tryptone, and (**D**) LB media with and without 1 mM IPTG supplementation. (**E**) The average mature biofilm of PSm was calculated from timepoints 8 to 24 h. The arithmetical averages are shown. Brackets show the 95% confidence intervals, and the homogenous groups are shown above the columns by the lower-case letters. Identical letters denote non-significant differences (*p* ≥ 0.05) between averages of biofilm.

**Figure 6 microorganisms-10-00618-f006:**
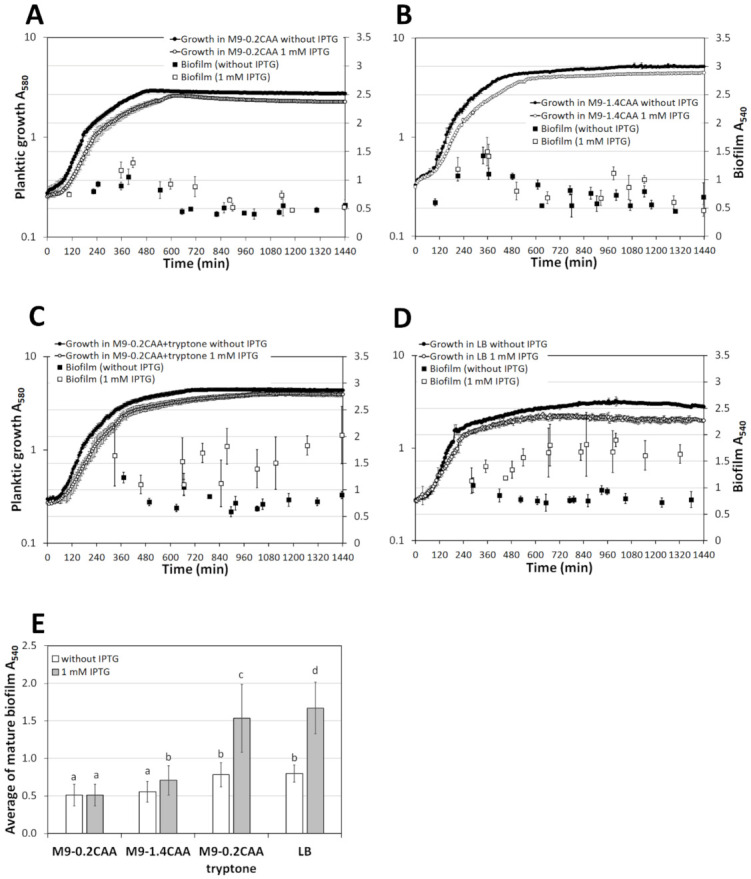
The biofilm dynamics and growth curves of *P. putida fis*-overexpression strain F15 grown in (**A**) M9-0.2CAA, (**B**) M9-1.4CAA, (**C**) M9-0.2CAA+tryptone, and (**D**) LB media in the presence or absence of 1 mM IPTG. (**E**) The average mature biofilm of PSm was calculated from timepoints 8 to 24 h. The arithmetical averages are shown. Brackets show the 95% confidence intervals, and the homogenous groups are shown above columns by the lower-case letters. Identical letters denote non-significant differences (*p* ≥ 0.05) between averages of biofilm.

**Figure 7 microorganisms-10-00618-f007:**
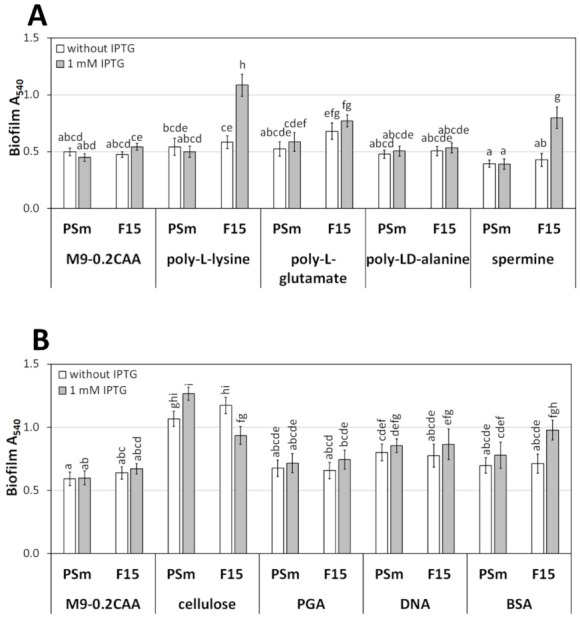
The biofilm of *P. putida* in M9-0.2CAA supplemented with biopolymers. (**A**) The biofilm of *P. putida* strains PSm and F15 in the M9-0.2CAA medium supplemented with 0.4 mg/mL of poly-L-lysine (pK), 0.4 mg/mL of poly-L-glutamate (pE), 1.4 mg/mL of poly-LD-alanine (pA), or 0.4 mg/mL of spermine. (**B**) The biofilm of *P. putida* in M9-0.2CAA supplemented with water-soluble Na-cellulose, poly-galacturonic acid (PGA), salmon sperm DNA, and bovine serum albumin (BSA). The arithmetical averages of at least three independent measurement sets are shown. Brackets show the 95% confidence intervals, and the homogenous groups are shown above columns by the lower-case letters. Identical letters denote non-significant differences (*p* ≥ 0.05) between averages of biofilm.

**Table 1 microorganisms-10-00618-t001:** The Spearman correlation analysis for growth parameters, biofilm, IPTG, and tryptone for *P. putida* strains PSm and F15. *P*-values are shown in brackets.

Strain		Growth Rate	maxOD	IPTG	Tryptone ^1^
PSm ^2^	biofilm ^3^	−0.526 (9.4 × 10^−6^)	0.547 (3.5 × 10^−6^)	0.084 (0.510)	0.873 (1.2 × 10^−20^)
	growth rate		−0.865 (6.7 × 10^−20^)	−0.154 (0.229)	−0.243 (0.055)
	maxOD			0.217 (0.088)	0.253 (0.045)
F15 ^2^	biofilm ^3^	−0.099 (0.476)	−0.081 (0.551)	0.306 (0.022)	0.873 (1.8 × 10^−18^)
	growth rate		−0.543 (2.2 × 10^−5^)	−0.378 (0.005)	−0.031 (0.823)
	maxOD			−0.415 (0.001)	0.051 (0.707)

^1^ The tryptone was defined as “1“ (LB and M9-tryptone) and “0“ (M9-0.2CAA and M9-1.4CAA). ^2^
*P. putida* strains, PSm and F15, were analysed separately. ^3^ The average value of every media biofilm was used for analysis as the number of measurements was different for biofilm and growth parameters of planktic cells.

## Data Availability

All data generated or analyzed during this study are included in this published article.

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
