# Peer review of "Tryptone in Growth Media Enhances Pseudomonas putida Biofilm"

_microorganisms, 2022, doi:10.3390/microorganisms10030618_

Round 1

Reviewer 1 Report

This manuscript studied biofilm formation by Pseudomonas putida. Different media were tested to see how nutrient components affect P. putida biofilms. By running a serious of biofilm experiments, positively charged components were identified as a biofilm stimulator on fis-dependent manner.

Title is too general and should be rephrased based on the findings.

Why were only three peptides tested? Nutrient rich media contain various amino acids and peptides and some may have positive and others may have negative effects on biofilm formation and other amino acids/peptides other than poly-L-lysine may affect biofilm formation.

L438: Please confirm nutrient media contain poly-L-lysine at the concentration of 0.4 mg/mL.

Author Response

Thank you for your review. 

1) The title is changed.

2) Yes, tryptone contains many different peptides, but we had to focus on the media components in general. The fractionation of tryptone peptides is a separate study, and we plan to do it in future. We studied the effect of different peptide mixtures on the biofilm in the form of peptone and tryptone (Fig. S2). And saw that peptone in growth medium enhances less biofilm. As pepsin yields peptides with hydrophobic amino acids at one end and trypsin produces peptides with positively charged amino acids, it encouraged us to ask whether the charge of the peptides causes an increased biofilm of P. putida. Additionally, we investigated the effect of donated peptides, the sequence of which we did not know, on biofilm. These peptides had a different impact on the biofilm; some had a positive effect, and some had no effect. However, we omitted these results from the manuscript as they would not have added much to the current research because we did not know the sequence of the peptides and could not repeat the experiment enough. Quantities were too small for sequencing and repeating the experiment.

3) Yes, we used poly-L-lysine at the 0.4 mg/mL concentration.

Reviewer 2 Report

This is a very interesting study, where Puhm et al., shows that the proteinaceous media components induce the biofilm formation in P. putida. Overall, this is a well-constructed study. However, I suggest the authors to address the following points.

Major Comments:

1) Page no 4-5, Results, Fig 1 and associated data;  LapA is the major player in the Fis mediated biofilm formation, therefore, I suggest the authors to include the LapA expression data of both wild type and F15 under both minimal and LB media conditions along with this panel. It is possible that the over expressed LapA simply facilitate adhesion to polystyrene surfaces, which is likely to be enhanced in the presence of catanionic peptides originated from tryptone. It is ideal that the authors provide the qPCR data for both LapA and Fis.

The data given on the Fig 1B suggests that the Fis undergo proteolytic cleavage. Interestingly, the cells grown in minimal media without tryptone shows intense band for low molecular weight fraction. It is presumable that the Fis undergoes enhanced proteolytic degradation in minimal media compared to LB or media supplied with tryptone. Therefore, I suggest the authors to do a quantitative analysis of these immunoblots and look for any potential correlations between the biofilm formation and LapA expression. It is possible that enhanced proteolytic degradation of the Fis reduces the expression of the LapA, therefore biofilm attachment.   

2) Instead of supplying the media with cationic or anionic peptides such as poly-lysine or poly-glutamate, the authors are suggested to deplete the peptide fractions through ion exchange chromatography as Turner et al., did in Antimicrob Agents Chemother 1998, 42; 2206–2214 (DOI: 10.1128/AAC.42.9.2206). The use of peptide such as poly-lysine could simply enhance over all cell binding to abiotic surfaces. Besides, these peptides can often activate two component systems such as PhoPQ, which often plays key role in the biofilm formation.

Minor Comments:

Page 6 lines 229-231: Please makes sure that the Fig panel referring in the text is correct.

Page 18 lines 548-549: There is no experimental evidence that suggest Tryptone is structural part of the biofilm, though it enhances biofilm formation.    

Author Response

Thank you for your review. We improved the style and grammar of the manuscript. 

1) Yes, this topic is exciting, and, as with research, many questions and hypotheses arise. However, to maintain compactness, we focused on media components' effect on the biofilm in this manuscript. It was important for us to rule out the possibility that the added tryptone only increases the biofilm as an additional source of energy and carbon. Indeed, we also concluded that lapA expression might affect biofilm formation in the presence of peptides (in the discussion section). In this manuscript, we decided to focus on the medium component that affects the biofilm and, in the second one, the association of lapA expression and LapA domains with peptide-forming biofilm formation. However, as always, dependence on peptides is more complex than it seems from first sight. The second manuscript has been written and is awaiting sent out.

We do not have a plausible qPCR reference standard because Fis increases stable RNA expression (and several common standards used for qPCR) in g-proteobacteria. Therefore, the accuracy of the qPCR results obtained from F15 grown with and without IPTG would be questionable. One can confirm, by β-galactosidase assay, the fis-overexpression enhanced lapA transcription similarly in all studied media (it will be referred to in the following manuscript).

Comparing the amount of Fis in media LB and M9-1.4CAA are similar in figure 1B, but fis-overexpression in M9-1.4CAA medium, like M9-0.2CAA, does not enhance biofilm (Figure 4). Thus, the Fis-enhanced biofilm in media supplemented with 1 mM IPTG does not appear to depend on the dosage of fis-overexpression (degradation), and fis is expressed in any case in excess.

We supplemented the manuscript with a remark. (lines 204-205; Track Changes with the option "All Markup")

2) Indeed, the question of which peptide of tryptone enhances the P. putida biofilm is very intriguing. We saw that tryptone and peptone had different effects on biofilm formation (Fig S2). Trypsin and pepsin produce peptides with different terminal charges. Moreover, there is no reference that adhesins of P. putida, like LapA and LapF, would have domains for specific peptides. Therefore we assumed that biofilm enhancement might happen via unspecific peptides (or with weakly conserved sequence), and the charge of the peptide is essential. We excluded from the manuscript the experiments with peptides with the unknown sequence that we obtained from colleagues because we could not sequence the peptides because of small quantities, and we did not have material for biological parallels. The effect of these peptides varied; some of them increased biofilm some of them had a neutral effect. Thus, in this case, the commercial peptides are reliable for preliminary study, the content of media is known, and the experiments are repeatable. The fractioning of broth to identify peptides that would increase the biofilm is a separate piece of research. We focused on the components of growth media in general in this manuscript and left the fractioning in the future.

If poly-lysine enhances overall cell binding to the abiotic surface, then poly-lysine would improve the binding of wild-type cells to the wall of wells. However, the positive effect of poly-lysine appeared only for F15 grown with IPTG but not for wild-type cells (Figure 7A). Thus, there must be a link between fis-overexpression and cationic peptides, and poly-lysine has to have a more specific effect on P. putida biofilm than unspecifically improved adhesion.

Page 6 lines 229-231: Text is corrected.

Page 18 lines 548-549:  The text is changed, and the conditional form is used.

Round 2

Reviewer 2 Report

I am satisfied with the authors comments, and looking forward to their manuscript on the role of LapA in media induced biofilm formation. 

Regarding the poly-lysine argument " However, the positive effect of poly-lysine appeared only for F15 grown with IPTG but not for wild-type cells (Figure 7A)." requires more experimental evidence. Poly-lysine is highly cationic peptide, for the above statement from authors to be valid, the authors need test the bactericidal effect of Poly-K used against both F15 and wildtype strains, with and without IPTG. It is possible that over expression of LapA ( if  there) sequester poly K from interacting with lipid membranes. They will also need to validate the cell viability in the observed biofilms using metabolic probes such as XTT or MTT or live dead staining. The presented data indicates only total biomass.